# Digital Technologies and the Transformation of Archaeological Labor

Eric E. Poehler 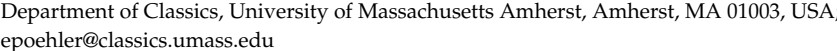

Department of Classics, University of Massachusetts Amherst, Amherst, MA 01003, USA;
epoehler@classics.umass.edu

**Abstract:** The use of computers and other digital technologies have had a long history in classical archaeology, but in the last decade, advances in software and especially hardware have begun to transform the way that archaeologists work in the field. This paper explores three examples of this phenomenon from my perspective as co-director, director, or assistant director of three different research projects between 2010 and 2019. These are the Pompeii Quadriporticus Project (2010–2013), the Pompeii Artistic Landscape Project (PALP, 2018–present), and the Tharros Archaeological Research Project (TARP, 2019–present). As a whole, these projects trace one history of digital technology's impact on the organization of archaeological labor, from intensifying work due to increased efficiency, to increasing the pressure due to newly available data sources, and to reorganizing the in-field procedures that at once takes advantage of efficiencies and frees up labor at the trench edge.

**Keywords:** labor; fieldwork; Pompeii; Tharros; workspace; digital; technology; efficiency; classical archaeology

## 1. Introduction

In addition to its physical tools, archaeology has a long tradition of borrowing its intellectual, organizational, and informational technologies. Even the shortest history of this borrowing would likely lead one first through the adoption of stratigraphic principles from geology in the 18th century, to their more fulsome articulation into archaeological practice in the 20th century [1], and to the further theorization of stratigraphy in the early 21st century [2]. This brief history might also introduce one to the concept of seriation and the biological models of evolution which inspired it [3,4] (pp. 75–80) or to newer bodies of spatial theory stemming from human geography, such as phenomenology and landscape archaeology [5]. Any such history should also include discussions of the military and industrial models that have been used to organize human labor in archaeology and to manage that labor for greater efficiency [6] (p. 422), [7] (pp. 424–426). Most recently, archaeologists have been adopting digital technologies eagerly, but the implications of this trend for the discipline are only beginning to be understood [8]. In this paper, I wish to explore these latter two elements—the organization of archaeological labor and the emergence of digital tools—in the context of three archaeological projects that I have been involved with as either a director, co-director, or assistant director, all of which have relied upon digital technologies to achieve their results. Two of these are field projects separated by a decade, while the other is an ongoing digital humanities project that involves organizing an equivalent amount of human labor. These three projects, two concerning Pompeii and another focused on the Romano-Punic site of Tharros in Sardinia, illustrate how digital technologies can be deployed to (1) manage and redistribute archaeological information in the field; (2) provide workspaces and training materials for undergraduates unfamiliar with certain concepts and data types in classical archaeology and art history; and finally, (3) to determine how new 3D technologies are re-organizing the shape of archaeological field projects, including the personnel they require and the moment-by-moment activities they engage in the field.

## 2. Pompeii Quadriporticus Project

The Pompeii Quadriporticus Project (PQP) was an archaeological and architectural research project (2010–2013) that focused on one of the largest and most important monumental buildings in the ancient city of Pompeii, Italy [9–12] (Figure 1). Combining new in-field information technologies, an exhaustive examination of the physical fabric of the Quadriporticus was conducted, and incorporating the results from an excavation running concurrently along its easternmost borders, the PQP endeavored to put this long-ignored monument back into its architectural and urbanistic contexts. The PQP overlapped its longer running sister project, the Pompeii Archaeological Research Project: Porta Stabia excavations, not only because Prof. Steven Ellis co-directed the PQP with me, but also because I was the head of architectural studies on those excavations [13]. This interdependent relationship meant that we could share resources and methodological advances across both projects. The primary method of analysis for the PQP, called masonry analysis, operates by identifying and distinguishing individual events of construction on the face of a wall—from the initial foundation to modern conservation efforts—and ordering those events into a relative sequence. The interpretation of a given wall's history is then combined with all the other walls of a structure to produce the larger history of a building and its environs [14].

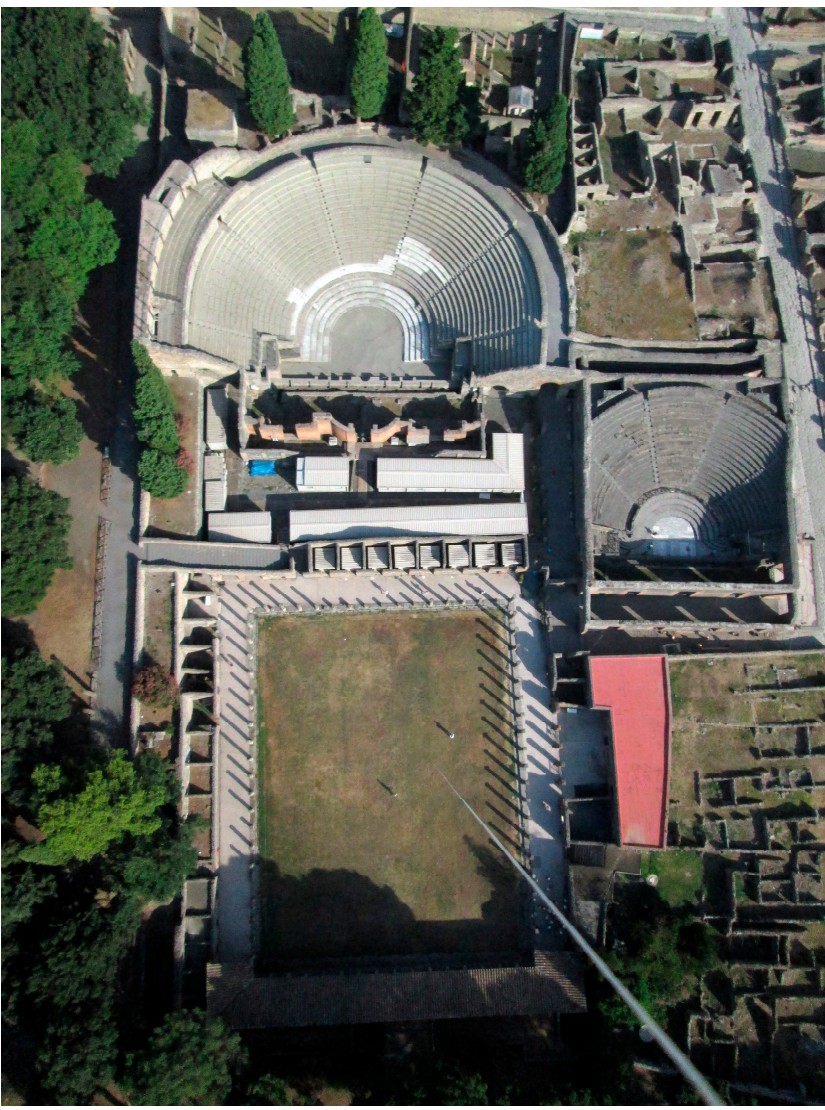

**Figure 1.** Overview image of the Quadriporticus, Pompeii.

The PQP was also a laboratory for the investigation of digital technologies in support of a noninvasive research approach. To document the three-dimensional space of the building, we used both a laser scanner and a few early commercial photogrammetry programs, including Autodesk's pilot program Project Photofly, which we continued to use as the cloud-based platform 123D Catch after that product's official release. In 2011 and 2012, a few other projects at Pompeii were attempting such near comprehensive use of photogrammetry. Other projects would soon use this technology more effectively and with better results [15]. With the demise of 123D Catch in 2017, we turned to Photoscan by Agisoft (rereleased as Metashape in 2019). To look below the ground in the open area of the Quadriporticus as well as within the surrounding colonnades and a few select rooms, we partnered with the British School at Rome to conduct two campaigns of Ground Penetrating Radar. The results of these surveys demonstrated that not only was the Quadriporticus the first building on this site, but also that it remained (with its internal alterations) the only construction here for approximately 240 years until the eruption of Mt Vesuvius. Additionally, these GPR results were important for tracing water infrastructure throughout the building and were especially valuable in identifying changes to that infrastructure in the post-earthquake(s) period.

The most influential technology that we brought into the field, however, was the tablet computer. Although now ubiquitous, the iPad was still in its infancy (indeed, only two months old) when we brought it into the field in June, 2010. As with photogrammetry, other archaeologists, far beyond Pompeii, soon also made great use of this device [16]. This device allowed us immediate access to two crucial technologies—a customized database and drafting applications—that we used to record our interpretations of the events of construction and reconstruction within the walls of the Quadriporticus. Our recording structure was now mobile and thus each student could document and organize observations in a single database, and, after each daily sync of all the iPads with one another, each student could access the interpretations and analysis of other walls within the building. This access permitted us to begin to see patterns within the data from disparate areas of the Quadriporticus at the same time that we standardized elements of description. Over the life of the project, the need to sync the data across all the iPads also brought new rhythms to our daily routine. At lunchtime, I collected, backed up, and synced all our devices so that the technology "up time" could be scheduled during the students' "down time". By 2013, however, the size of our team and the number of our iPads had grown to twelve and a lunch hour was insufficient time to run the sync process. This additional sync time made it necessary to plan out how to prioritize the work device of one student over that of another and tethered one of the directors to a computer and stack of iPads for as much as 20% of a work week. In this sense, the labor costs associated with the iPad rolled "uphill".

Offsetting this cost to the senior staff was the efficiency offered by the drafting applications. With these apps, our team took a picture of the wall being analyzed and then, using different layers, drew over that wall, tracing the outlines of the constructions and reconstructions they had identified through masonry analysis. Drawing on the tablet both produced high-quality illustrations of each wall and did so at a rate significantly greater than working with paper-based drawing. In traditional paper-based drawings, it was necessary to set up a baseline across the wall using a level string and tape measure and to make measurements along that baseline, documenting the location of each of the different construction events identified in the wall. Depending on the size of the wall, this might take a few hours to more than a full day. Drawing the same wall on a tablet, however, could be accomplished, after some practice, in half an hour. This anecdotal assertion is supported by our calculations of work rates for the first two seasons, when the project went from one iPad for every two people (2010) to supplying one tablet for each member of the team (2011). Without the bottleneck created by a limited number of tablets, the PQP team completed 371% more work with 35% fewer people [10].

Moreover, the result of this drafting process was a kind of double product, including both the abstract drawing of our stratigraphic units as well as an image of the underlying

wall. Of course, using a picture of a wall to draw over it meant also that we must have an image for this drafting process to work. On occasion, capturing or creating those images required its own workflows and labor costs. For example, for those walls that we could not fully capture in a single image (e.g., walls we could not get far enough away from to capture the wall in a single image or, conversely, very large walls that we could not capture sufficient detail in a single image), we needed to stitch together multiple images in photoimaging software. This both delayed the drafting process and introduced uncertainty in the image accuracy, as the process of merging the images also had to mathematically compensate for the different angles of each image. Additionally, it was necessary in the 2010 and 2011 field seasons for us to take images with a digital camera and then, using a SD card reader accessory, upload the required images onto the iPad. Only in 2012, with the release of the third generation iPad and its 5 megapixel camera, was it possible to subsume image capture within the drafting process. This hardware advance led to a more streamlined workflow and thus to further efficiency within the project.

The time afforded by this efficiency was refocused in a number of additional projects that expanded the possibilities for on-site research into areas more traditionally performed in the off-season. The first of these additional projects brought more than 140 paintings, drawings, and sketches from the 18th century onward into the field and thus into closer dialog with the architecture. By loading these archival materials onto our iPads, we could often stand in the very place within the building where the artist stood (or imagined he stood) and consider the differences between the art historical record and the material remains before us. Although artistic license and simple error were a constant concern, the comparison with these materials provided evidence to judge which reconstructions of the masonry were created in the early modern period and which aspects of the building were the final, ancient additions to the Quadriporticus. Most importantly, this research produced an opportunity to connect a few small details within a few images spanning half a century (1770s–1820s) to the recent images from the GPR surveys. Together, these images, separated by centuries, revealed that a circular, colonnaded structure once stood (or was being constructed) in the center of the courtyard that had never been recognized previously [17].

A second investigation taken up on the back of the efficiency in our digital drawing techniques was an intensive analysis of the interior colonnade's 74 columns as well as three more columns from the propylon. At the heart of this study were a series of measurements that we took on each column, including (1) the height of each column drum, (2) the position of that drum in the column (from bottom up), (3) the height of the plaster jacket where it existed, (4) the location and description of other plaster that remained on the column; (5) the location and description of damage to the column, and (6) the location and description of "interventions" on the columns, such as holes and cuttings. We also recorded related changes to the stylobate around the columns and in the intercolumniations. The result of these efforts was a database of hundreds of measurements and observations concerning the 240 year history of a continuous colonnade. Some of these analyses are still on-going, and so I will focus on a subset of the 1732 observations to illustrate how the iPad offered more than mere efficiency in wall drawings, but also how it supported our development of novel methodologies and the related workspaces while in the field.

Early on in the project, we noticed that some number of cuttings in the columns seemed to face across the intercommunication to a responding hole in an adjacent column. As a result of the data we had collected, it became possible to explore this observation more explicitly and more exhaustively. To do this, I built a layout in our Filemaker database that showed the data about a particular intervention from a single column on one side of the screen, while the other side listed the data of all the interventions of a neighboring column (Figure 2). Color coding was used to help discover matches, for example, in locating the position of an intervention around the circumference of a column. We used the metaphor of a clock face to establish that position, which meant that any intervention on the north–south running colonnade that faced another column would be recorded as

6 o'clock or 12 o'clock, while those that faced onto the building facade or into the open courtyard would face "3 o'clock" or "9 o'clock". These positions were reversed for the east–west running colonnades. The color given to each of these positions helped the student to visually sort through the positional data and narrow the possible number of matching interventions. Next, the height data about each intervention were listed, also coded by color into bands of one meter. With the position established, the student then looked to see if an intervention was at an approximately equal height (our tolerance for a match was ca. 5 cm) and could have made a relatively level connection between the columns.

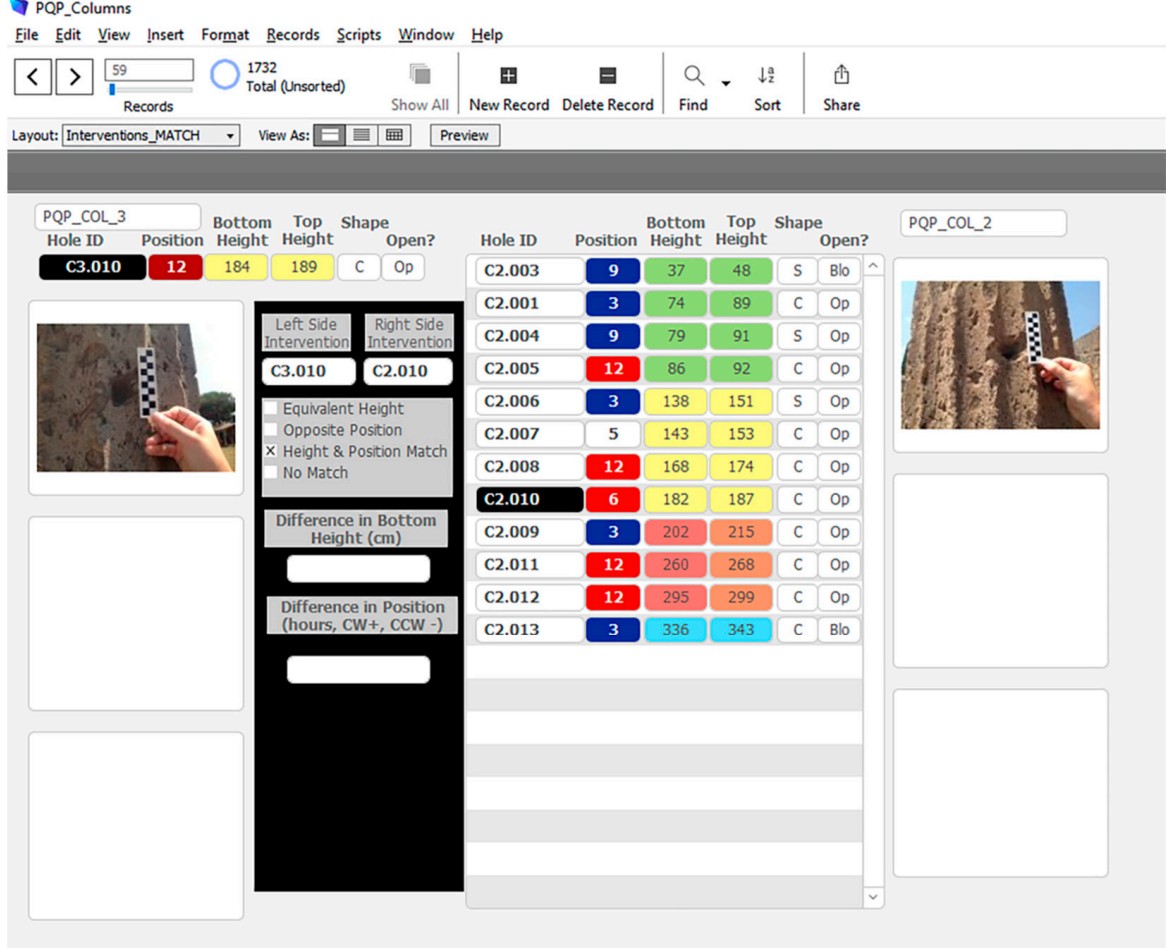

**Figure 2.** Interface of FileMaker database for analyzing interventions in the Quadriporticus colonnade.

Although later constructions, especially the plastering over of earlier interventions, have made a complete reconstruction of features placed between columns impossible, the information in the aggregate remains especially revealing (Figure 3). If we take only the number of interventions for each column within the eastern colonnade that (1) face another column (blue), (2) that face onto the interior of the colonnade (red), and (3) that face into the courtyard, we see a remarkable pattern appear. At both ends of the colonnade, the number of interventions is particularly high, both internally and between columns. Moving towards the center from each end, however, we see the number of interventions dip before increasing again only to fall as they meet the center intercolumniation. It is here, at this middle point of the colonnade, that cuttings in the stylobate suggest a framework had been inserted to form a doorway, accessing the center of the Quadriporticus' open area. Moreover, the number of interventions between columns on either side of this portal suggests that what might appear to be one of the most permeable forms of architecture was in fact not so. Rather, movement through the columns was directed to a single point and

the decorative elements among the columns—which we might reconstruct from analogy with examples of wall paintings as crossbars that supported shields and masks, garlands, lattice-work fences, and statue bases—were more than merely ornamental, serving to organize human movement as well.

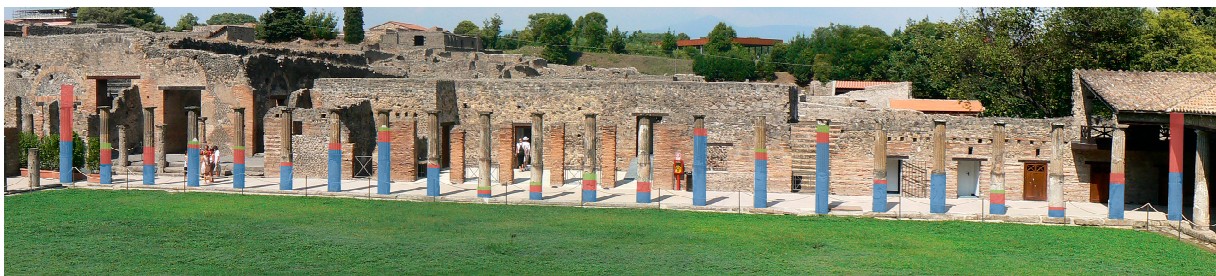

**Figure 3.** Eastern colonnade of the Quadriporticus with overlay of relative intensity of interventions facing another column (red), facing the building façade (blue), and facing the open area (green).

What I hope the preceding paragraphs demonstrate is that intriguing and novel results not only stem from interesting and novel digital work practices, but also from the efficiency of work afforded by these technologies. In the PQP, it was immediately clear that our increasing work rate would lead to additional time on site, and that time offered a choice about how to deploy it. We were primarily interested in using that opportunity to increase the amount of effort we spent interpreting our results, both in terms of aggregating individual observations among stratigraphic units of adjacent walls and in making comparisons across the building as a whole. This led us to seek a better understanding of how the long history of artistic documentation of the building could be used to disentangle the many reconstructions present in the masonry. Similarly, the time offered from technological efficiency also encouraged us to make a detailed archaeological, rather than architecture historical, analysis of the building's colonnade. This work created a remarkable new dataset and required the development of new methods and new digital mechanisms to understand, analyze, and re-combine those data into a new history of an old architectural form.

### 3. The Pompeii Artistic Landscape Project

Efficiency, of course, is not a goal in its own right, nor should it simply be used as a way to maximize archaeological labor by creating the conditions in which more of the same work can be performed by fewer humans or in which the amount of critical thinking or human engagement in the process is reduced. One must actively avoid creating systems that make humans mere automatons within a larger push-button structure, and instead we should build digital systems that layer new forms of expertise and their learning curves on top of already important domain specific knowledge and skill sets. In the PQP, we did this (to the degree that we were able) by first having an established set of practices on how to perform masonry analysis already in place from the non-digital world. This meant we largely translated practice from one analog universe into the new digital sphere. For the Pompeii Artistic Landscape Project, a necessary reliance on untrained and under-trained undergraduate students has meant that the digital framework we built to organize their labor needs to provide not only a workspace, but one that leverages a familiar set of tools and provides an array of custom support files for their reference.

The Pompeii Artistic Landscape Project (PALP), generously funded by the Getty Foundation, grew out of an earlier project, the Pompeii Bibliography and Mapping Project, which has provided much of the underlying spatial data that allow us to place the art historical data in very specific locations within Pompeii. I will return to the implications of the PBMP briefly at the end of this section. PALP is a project based on the observation that Pompeii provides the single largest continuous art historical landscape from the ancient world. We see the ancient world speaking to us through these artworks and

revealing insights onto innumerable aspects of Roman life: Roman appetites for food, sex, and mythologies; their identities as revelers, intellectuals, and professionals; and their practices of religion, politics, idyll, and idle. Moreover, the scale of the city means that there is a continuous and contemporaneous landscape of the Roman imagination more than 400,000 m$^2$ in area to be explored and considered. Moreover, because it has been studied for over two and a half centuries, Pompeii possesses a massive bibliography and a robust digital infrastructure, both of which are essential to PALPs success. For example, from the PBMP, there are detailed digital representations that telescope down from the nine regions to the more than 110 city blocks, to the 1100 buildings, to the approximately 10,000 rooms within those buildings, and then to the approximately 81,000 walls that surround those rooms. There are also two crucial sets of catalogs of the artworks in Pompeii. The first is the four volumes of Pompei. Pitture e Pavimenti (PPP), which published efficient textual descriptions of the style, condition, and documentation of each artwork. The second catalog is the eleven volumes of Pompei. Pitture e Mosaici (PPM), which offers more fulsome descriptions of the content of the artworks in captions that accompany thousands of photographs and drawings. Finally, PALP has been given access to nearly 60,000 high resolution images from the website Pompeii in Pictures. These images offer more detail and more context for the artworks, and most importantly, they show the color present in the artworks, as the PPM volumes are published mostly in black and white.

As in other databases [18], these data, however grand in scale, still need additional enrichment, description, or contextualization. For example, in order to fully embed our different representations of space (e.g., buildings, rooms, and walls) within the larger context of the city, we needed to apply geoprocessing tools so that each room "knows" what building it is in and each wall "knows" what room (or street) it faces onto. For the Pompei. Pitture e Pavimenti (PPP) and the Pompei. Pitture e Mosaici (PPM) volumes [19,20], the issue was initially one of extraction—getting the thousands of text entries (PPP) and images and caption sets (PPM) from the printed books into our digital workspace. Even after these materials were extracted, there were additional barriers for our undergraduate workers to overcome within the content itself; the descriptions are written in Italian and replete with unfamiliar art historical terms that, even when translated, do not always make sense in common English. Finally, there was the additional problem that even if our students were domain experts and fluent in Italian, they still did not have experience in the linked open data formats into which we intend to redescribe all these artworks. Indeed, a survey of first year students in my "Digital Tools and the Academy" course at the University of Massachusetts Amherst showed a surprisingly low digital literacy even among those who self-selected for this course [21].

That same survey also showed that all students had both experience with and comfort in using Google products such as Docs, Sheets, and Slides. Since, at its most basic level, our work is to transform information encoded in images and in narrative texts into a more machine-readable format, the students' familiarity with Google Sheets made it an obvious technology to underpin a student workspace. Being situated in a web browser rather than in a downloaded program such as MS Excel means that every student has access to the technology (and the same version) and that each workspace can be edited online, shared with other students and staff, and stored securely. For these reasons, PALPs first year of work focused on building a workspace that met the needs of our workforce, one that provides not only a simplified workflow to transform information from one structure (e.g., image and narrative) to another (linked open data), but also help students overcome the domain-specific knowledge they did not possess at the beginning of the project. We built our workspace, over many iterations, as a sequence of tabs within a webpage that structured the use PPP data into one tab, access to image data in another, and an environment for description in the third. With all available data at hand, students can redescribe the artworks in the description tab, identifying each motif mentioned in the texts and adding other depictions visible in the images but not otherwise described. Students do this by selecting from a series of categories (e.g., figures, objects, architectures, animals, plants, and motifs)

that link to a dropdown accessing a controlled vocabulary of depictions. Additionally, students select the spatial position of the artwork (e.g., socle, central panel, and lunette) and any attributes those depictions or spatial elements might possess or display (e.g., color, action, and multiples) (Figure 4).

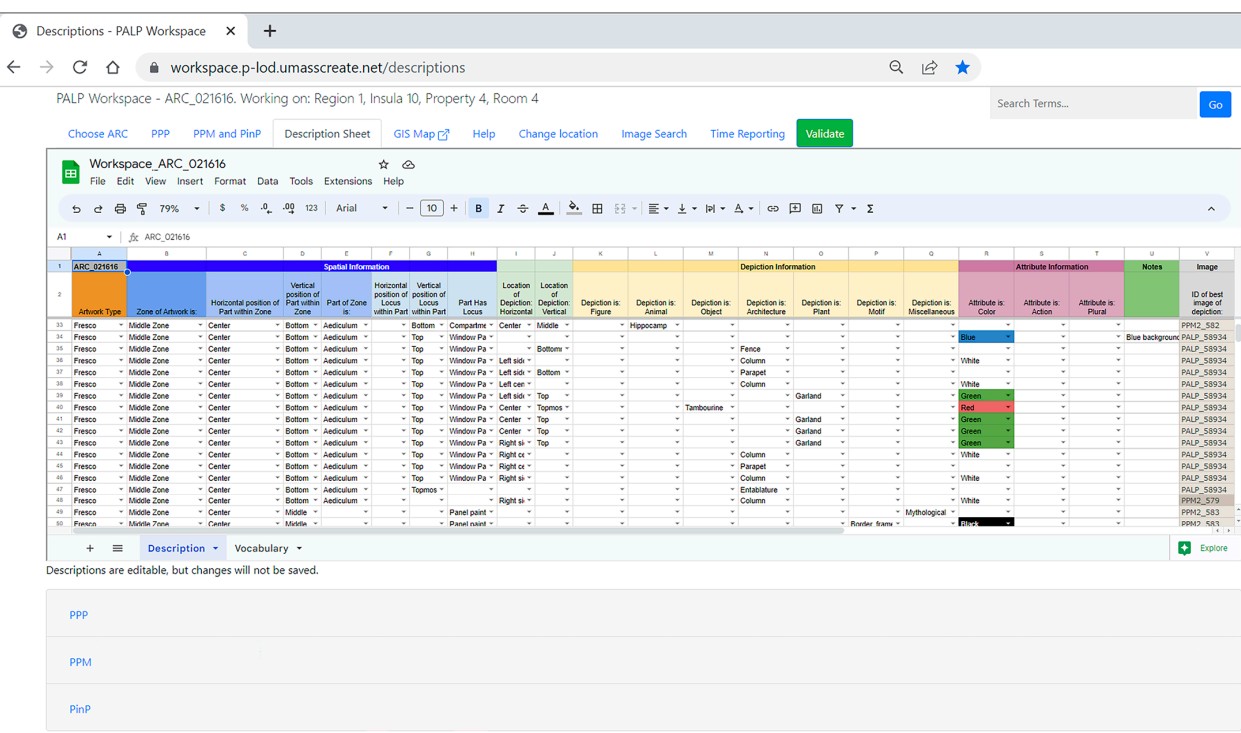

**Figure 4.** Example of student workspace for describing artworks in the Pompeii Artistic Landscape Project.

Our browser-based workspace offered additional tools, such as automated translation from Italian to English and outlinks to other sources of information, including reference data from the Getty Art and Architecture Thesaurus and our own tutorials and vocabulary definitions. These were essential tools in unlocking the information in the Pompeii. Pitture e Pavimenti and Pompei. Pitture e Mosaici volumes for students to use, instantly transforming a foreign language of technical terms into rough English with an accompanying dictionary to look up those translated terms. These tools, however, were not enough and it has been necessary to build our own help instructions and files to explain the work and to clarify the relationship between the data, the workspace, and the purpose of the work. For example, in the header rows of the workspace sheets (Figure 4), we color coded columns to show the progression of description from the Artwork type (orange) to larger Zones of a fresco (dark blue) to the Parts (light blue) that can divide up that zone, and from the Loci (lightest blue) that can further subdivide that Part to the specific depictions (yellow) that might be found in those locations and the attributes such depictions might have (mauve). Additionally, to know what any of the (currently) 555 depictions are, we provided students with a term search tool (i.e., type a term and see a definition and an image) as well as a visual search platform (i.e., scroll and click through images to find a term and definition). Finally, we have created a series of video tutorials, a reference collection of complex artworks and examples of their description, and a searchable "bank" of feedback given to students as their descriptions were corrected.

The preceding summary of the Pompeii Artistic Landscape Project's workspace, however compressed, is sufficient to illustrate how digital archaeology and digital art history projects produce particular digital labor conditions and require solutions specific to its digital milieu. This means acknowledging and appreciating what digital and domain skills

and familiarities students have, those they do not have, and what solutions are possible to build in the digital realm to address those realities. Less explicit in the preceding paragraphs, but no less crucial, are the organizational skills, flexibility, and simple salesmanship required of the directors of a digital project to recruit and train a small army of students and to communicate and coordinate with faculty, administrators, and staff in disciplines far outside their area of expertise (especially computer science and data science). It is perhaps useful to make the comparison of directing a digital research project with the more familiar role of directing a traditional archaeological field project, particularly in the administrative burdens required to get a project going, keep it running, and see it to publication. Where these differ, however, is in the specialized digital skills and knowledge (and often their invention) that are required on top of the traditional archaeological methods and abilities. Of course, as the earlier discussion of the Pompeii Quadriporticus Project shows, field projects are now commonly digitally infused, and this new layer of digital labor is simply part of the bargain in exchange for the new possibilities offered in the digital world.

Finally, it is worth taking a moment to think through the implications for archaeological labor if PALP and its sister project, the PBMP, are successful in providing powerful tools for archaeological research. Let us imagine an excavation in Pompeii open in the year 2030 in which the excavator finds a demolished kiln and large fragments of wall plaster with mythological figures and distinctive motifs. In current practice, those finds would be recorded, described, and stored for more intensive study by specialists some time later. With the resources of PALP and PBMP available at the moment of discovery, there will be a temptation at the moment of discovery to engage with the secondary source material to understand these finds at the expense of the on-going excavation. In a perfect world, this would be a good thing, pausing digging for a moment to put the artefacts of an excavation into context in real time. However, time is the scarcest resource on any field project, and it is likely that in the real world, the time it takes to do this secondary source research will not come at the expense of digging and recording during the day, but at the expense of eating and sleeping by the trench supervisor at night. In this scenario, looking for bibliography, old maps, and artistic comparanda becomes as much a new burden as a new opportunity for the "middle management" of an excavation, and project directors must be aware of how new technologies will impact the work loads of excavation teams [17] (p. 215).

## 4. The Tharros Archaeological Research Project

One example of an archaeological project attempting to adjust the way it deploys its human resources in the face of a new, disruptive technology is the Tharros Archaeological Research Project (TARP), directed by Steven Ellis at the University of Cincinnati. I serve as the assistant director for TARP and am specifically responsible for the geospatial and 3D modeling techniques used in our digital recording systems. The broadest goal of TARP is to develop a clearer understanding of the city's urban development, which has until now been either only vaguely understood or, where is more confidently attested, often built upon outdated or incomplete bodies of data. Through a more critical and evidence-based delineation of urban development, particularly one that can be anchored to chronological markers recovered via excavation, our project seeks to connect the city's periodic urban expansions to broader aspects of Mediterranean history [22].

Informed and enthused by our experiences with the Pompeii Quadriporticus Project and its sister excavation, the Pompeii Archaeological Research Project: Porta Stabia, Ellis and I knew from the earliest planning stages for TARP that we would use both tablet computers for digital recording of our trenches and photogrammetry to capture the richest image of each stratigraphic unit and features encountered. For example, the experience of taking a database designed as a field recording tool on a tablet and adapting it to become a reporting instrument in support of the post-excavation publication efforts led us not only to define new categories of information to be recorded while still in the trench, but also to reorganize how information about soils, artefacts, and architectures flowed among the teams responsible for these areas. Similarly, our experiments with photogrammetry

convinced us of a need for a more comprehensive use of 3D models and for the development of a method for their capture that is more integrated with our daily practice. Fortunately, half a decade after the PQP ended, the practice of producing, sharing, and annotating 3D models has become far less specialized and more commonly embedded in archaeological practice [23,24]. In part, this is because basic computer hardware has caught up to the needs of the software, which, in turn, have improved their algorithms even as digital cameras produce images of ever higher resolution. Additionally, commercial hosting solutions, such as Sketchfab and p3d.in, have made sharing models simple and made enriching those models with annotated information possible.

With all these advances available, we conceived (as others have done similarly and before us) a process to produce a three-dimensional model of every stratigraphic unit excavated and to use the resulting imagery to help us create a traditional two-dimensional drawing of that stratigraphic unit in GIS. The first step in our "3D-to-GIS pipeline" was to set out between 15 and 20 targets around each trench to allow for automatic recognition of locations inside each photogrammetrical scene (Figure 5). Using a combination of total station surveys and drone reconnaissance, we derived longitude and latitude coordinates for four targets at each trench, which were subsequently used in the production of each 3D model. These models, therefore, were geolocated in real-world space, a feature that extended to the overhead orthomosaic images that we exported for use in GIS. As a result of these coordinates, each orthomosaic for a trench layered directly over the last in our GIS and allowed us to draw a representation of each stratigraphic unit and have that representation already positioned where it once existed in the real world. A final step in our "pipeline" was to export the polygons of each SU into our database, where the relative shape and position of each stratigraphic unit could be accessed by every member of the project. This feature is particularly useful for other trenches to compare their stratigraphy and for the artifact team to have a clearer sense of the context of the assemblages they are processing.

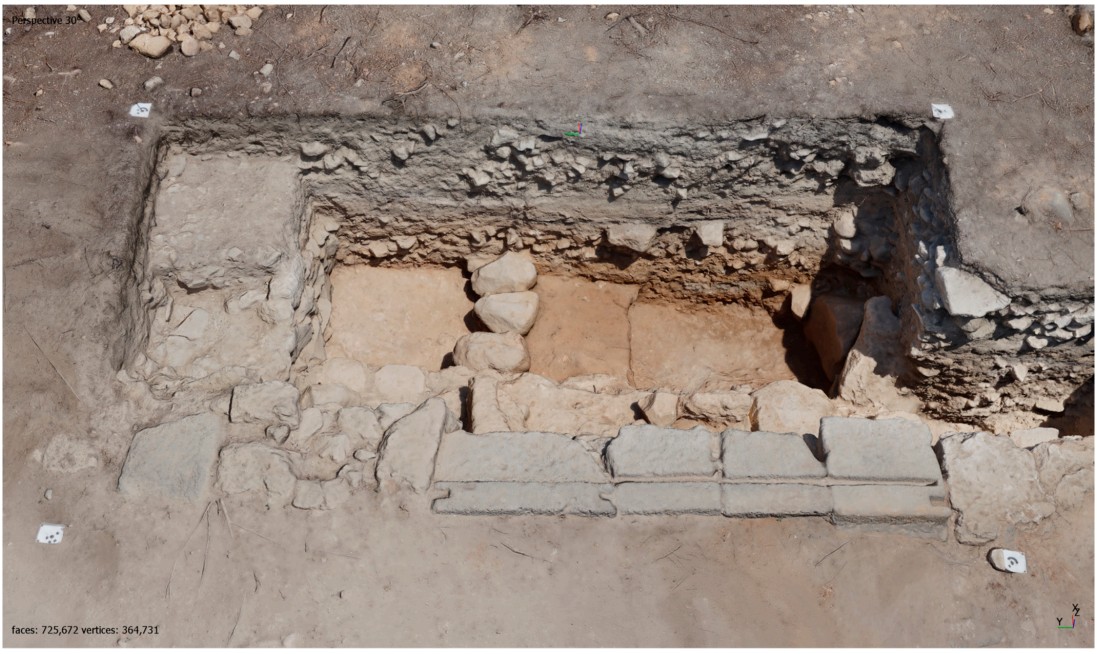

**Figure 5.** Three-dimensional model of Study Area 2000 at the Tharros Archaeological Research Project.

The most interesting and most consequential impact of introducing comprehensive 3D modeling into fieldwork is that the labor associated with it, at least in our design, shifts responsibility from the excavators (who previously were tasked with drawing features in the trench, whether on paper or into a tablet) onto specialized personnel who now shepherd all the drafting data through the "3D-to-GIS pipeline". Most importantly, this

requires the team excavating the trench to clean the area, remove the tools, and vacate the trench for 20 to 30 min while image capture is accomplished and a basic model is produced to assess the quality of the capture. Just as in traditional drafting, one cannot dig the next area until the last one has been fully recorded. Unlike traditional drafting, however, which shifted the entire team's focus of activity from digging to drawing, the vast majority of the labor associated with an excavation is now excused from the trench while the new drawing procedure and its dedicated personnel takes over the space. Assuming only one stratigraphic unit is recorded per day, this new procedure will create a minimum of between 33 and 50 person hours of idle time for a team of four over the course of a five week season.

It was immediately clear based on our previous experience with the projects described above that this change represented an opportunity to reorganize how we deployed the labor associated with the efficiency of recording in the field. Similar to the PQP, we chose to dedicate much of this time to tasks related to data management as well as beginning the interpretation and analysis of those data. For example, during the times of photogrammetric capture, excavators returned to each stratigraphic unit excavated that day or in previous days to ensure the quality of our data. We also used this time, most often by the supervisor, to assess how the new stratigraphic units fit into the interpretation built in previous days. Additionally, during the time the computer was processing images, other tasks were possible (since excavators could be in the trench but not digging), such as taking elevations and other measurements, sketching features, or taking photographs. Of course, fieldwork is a messy business, and all of this work out of the trench occurs in a more ad hoc manner than this short description suggests.

What is undoubtedly the most interesting implication for archaeological labor to come out of this drawing procedure was the recognition of the paradoxical relationship between the dynamism, richness, and immersion of the 3D model and the flatness, two-dimensionality, and opacity of the stratigraphic units drawn from those models. Simply put, the end product of our pipeline is far less impressive than the source data from which those drawings are created. As an analogy, it is like growing an apple tree and then only using pictures of apples grown. As a means of communicating information, however, two-dimensional plans have clear advantages in being widely understood within the discipline (and beyond) and in being broadly acceptable in most means of scholarly communication. Although these trench-side technologies are pushing ahead of the mechanisms required for sharing the information that they produce, there are still lessons to learn and progress, if more limited, to make. That is, while it might not be possible for an excavation to change disciplinary expectations for archaeological illustration or to push scholarly presses to accommodate 3D models in their publications, it is possible to continue to develop how 3D models are used as part of best practices in the field. Therefore, the question for TARP now is how we utilize the fullest potential of these new, data-rich, and visually engaging objects to better understand what we discover of Tharros.

Our answer—one of many possible answers—is to develop our 3D models as a mode of communication within the project and as a means of analyzing, describing, and sharing the information associated with and embedded in these 3D models. To accomplish this will require dedicated time and effort from the leadership and an investment in training and infrastructure across the project. Thus, in order to endow these models with information from the trench, we will need to train our excavators, at least the supervisors, on how to make this embedding of information possible. In order to share this information across the project, we will need a computer infrastructure of hardware and software that makes the movement of those models between devices and people as frictionless as possible. Finally, and indeed most importantly, we will need to invest in the training of individuals on how to simply "read" a 3D model. That is, students do not necessarily arrive in the field with an understanding of how an archaeological plan is produced or how it is read as a source of information once it is produced. There is no reason to believe that the vividness of a 3D

model will allow one to understand its archaeological and informational value without a similar amount of training.

## 5. Conclusions

Even when seen only through one person's experience of three interrelated projects, the impact of digital technologies on the organization of archaeological labor is clearly profound and multifarious. Indeed, if we examine only one aspect of that impact, the increased efficiency of fieldwork, one may fairly describe efficiency as beneficial and deleterious (Caraher 2016 [7]). The specific case studies in this paper, however, have focused more upon a few novel approaches that the additional time gained from such efficiency might be redirected to, such as the study of the Quadriporticus' colonnade or its archival materials. At the same time, the Tharros Archaeological Research Project shows how the deep investment in digital technologies can disrupt traditional workflows (i.e., drafting procedures) and lead to either significant inefficiencies or to new opportunities. Thus, TARP is reorganizing the labor idled by our new 3D recording process for interpretive tasks rather than for additional excavation in another trench.

Finally, these three projects all show that not only are digital technologies, such as databases and photogrammetry software, replacing analog recording materials and procedures, but also that these new technologies are increasingly becoming the intermediating frameworks between ourselves and our objects of study. Similar to the question of efficiency, and bound up with it, this issue of digital workspaces is both a concern and a necessity. Although categorically no different than paper context sheets, the data entry form on a tablet presents a highly structured and thus significantly limited means of interaction with archaeological materials. The carefully organized blank fields allow for easy capture of relevant information, but equally easily imply that if evidence cannot be described in this form, then it is not relevant. On the other hand, the evolving workspace of the Pompeii Artistic Landscape Project demonstrates that the simplicity of an interface (i.e., a Google Sheet) can disguise the complexity of the underlying information and its organization, allowing students without significant domain knowledge to make significant contributions to a research project. In this example, the intermediating structure of the workspace stands in for a too steep art historical learning curve and allows the student to apply the abilities that they have and that the computer as yet does not: critical thinking and creativity.

**Funding:** This research was funded multiple institutions and agencies. The Pompeii Quadriporticus Project was funded by Cardinal Intellectual Properties, the Five Colleges Inc., the Mellon Foundation, the University of Massachusetts Amherst, and the University of Cincinnati. The Pompeii Bibliography and Mapping Project and the Pompeii Artistic Landscape Project were funded by the American Council of Learned Societies, the Getty Foundation, the National Endowment for the Humanities, and the University of Massachusetts Amherst. The Tharros Archaeological Research Project is funded by the University of Cincinnati.

**Data Availability Statement:** Data from the Pompeii Quadriporticus Project are not yet available. Data from the Pompeii Artistic Landscape Project are currently searchable via the project interface (under-development) at http://palp.art. Three-dimensional models from the Tharros Archaeological Research Project are available at https://sketchfab.com/uctharros.

**Conflicts of Interest:** The authors declare no conflict of interest.

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
