# Peer review of "Digital Technologies and the Transformation of Archaeological Labor"

_heritage, doi:10.3390/heritage6050209_

Round 1
Reviewer 1 Report
The use of digital technologies in archaeology is now a very topical debate that has been investigated for years. The description of the three projects should be divided into sub-sections, better emphasising the different methodological approaches, the different resources used and the different results obtained. The topic is not innovative, as are the approaches implemented. The iconographic apparatus is scarce and should be increased to demonstrate the results achieved by the three projects, which result described in general. The bibliography should be updated, moreover 7 out of 19 bibliographical references are by the author. A comparison with more current projects should also be made in the conclusions; a state of the art in the present day is lacking. I absolutely agree that digital technologies cannot substitute the importance of the archaeological research approach and knowledge, but they can certainly help to achieve more accurate results in a shorter time, although with greater resources. Today, the approach to this type of project must be absolutely interdisciplinary.
Author Response
Reviewer 1 is correct in many aspects of their review and in the context of a single, new submission to a journal I would agree with the appropriateness of these remarks, especially the over citation of my own work. The context of this submission, however, is a long-delayed publication of the proceedings of the On Shifting Ground conference in 2019. My paper reflects my remit in 1. presenting my remarks, 2. the time in which they were written, and 3. the purpose of memorializing them in this paper. Thus, while I believe in a more neutral context, R1’s recommendation are appropriate, in this context they would undermine the purpose of the special issue in Heritage. There are bibliographic suggestions from R2 that I will incorporate.
[1]= "Because the context of this submission is a long-delayed publication of the proceedings of the On Shifting Ground conference in 2019, my paper reflects my remit in 1. presenting my remarks, 2. the time in which they were written, and 3. the purpose of memorializing them in this paper. "
Reviewer 1:
"The description of the three projects should be divided into sub-sections, better emphasising the different methodological approaches, the different resources used and the different results obtained." - I will not be changing the text on R1's point because of [1].
"The iconographic apparatus is scarce and should be increased to demonstrate the results achieved by the three projects, which result described in general." - I think R1 means to describe the number of illustrations by the phrase "iconographic apparatus". I will not be increasing the number of images.
"The bibliography should be updated, moreover 7 out of 19 bibliographical references are by the author." - This was also pointed out by R2 and several new citations have been added to the revised submission.
Reviewer 2 Report
The paper reports three interesting instances, from research projects in Pompeii and Sardinia, of how the analytical and digital technologies are increasingly supporting the research in the field of classical archaeology. The article is well written and provides an overview of the application in archaeology of different digital tools, i.e., on-field Pad-based image acquisition and processing, creation of digital catalogues, application of photogrammetry to archaeological excavations.
My concern mainly regards the weak bibliographic apparatus, that should be reinforced, as most part of the applications of digital tools mentioned in the text are already widely developed and commonly applied in their broad outlines to diverse archaeological projects.
In detail:
Topic 1 - PQP project
Lines: 58-61: I would mention how the problem of deciphering the ancient constructive phases and modern restorations of Pompeian buildings is particularly complicated. The current approached also integrates analytical techniques different from digital-based ones, as archaeometric investigations of building materials. For example, the study of mortars for scanning building interventions is offering important advances. I would add some lines on this topic, reporting some suggested references as:
- Dilaria et al. 2022a, Dilaria S., Previato C., Secco M., Busana M.S., Bonetto J., Cappellato J., Ricci G., Artioli G., Tan P., Phasing the history of ancient buildings through PCA on mortars' mineralogical profiles: the example of the Sarno Baths (Pompeii), Archaeometry 64, 4, 866-882.
- Miriello et al. 2010, Miriello D., Barca D., Bloise A., Ciarallo A., Crisci G.M., De Rose T., Gattuso C., Gazineo F., La Russa M.F., Characterisation of archaeological mortars from Pompeii (Campa-nia, Italy) and identification of construction phases by compositional data analysis, JASc, 37, 9, 2207-2223.
I would suggest, therefore, to mention the image-based working tool reported for PQP as one possible method supporting the traditional archaeological research and the investigation of construction techniques and materials.
Moreover, the applications of laser scanners, geophysics, and 3D-GIS for the analysis of masonries is becoming very common, with several applications also in Pompei that I suggest:
- Landeschi et al. 2015, Enhanced 3D-GIS: Documenting Insula V 1 in Pompeii, in CAA2014 21st Century Archaeology Concepts, methods and tools, Proceedings of the 42nd Annual Conference on Computer Applications and Quantitative Methods in Archaeology, Oxford.
- Santoro S. (Ed.), Pompei. Insula del Centenario (IX, 8) I. Indagini diagnostiche, geofisiche e analisi archeometriche. Studi e Scavi. Nuova serie 16, Bologna.
Lines 80-81: On field-iPad application. The application of iPad in archaeological research is not completely unusual. I would add reference to other instances of use of these devices in the framework of archaeological excavations, for examples:
- Berggren et al. 2015, Revisiting reflexive archaeology at Çatalhöyük: integrating digital and 3D technologies at the trowel's edge, Antiquity, 89(344).
Topic 2 - PALP project
The creation of digital on-line databases collecting all the information on classical artworks is extremely common, with many applications for cataloguing floors, mosaics and wall decorations. The databases are usually structured in queryable fields where all the information regarding an artifact is provided. We recommend viewing and citing, for example:
- Ghedini F., Rinaldi F., Kirschner P., Tognon M. 2007, TESS. THE ON-LINE DATA BANK OF MOSAIC COVERINGS, Archaeology and Computers 18, 2007, 13-43
- M. Salvadori, D. Scagliarini (eds.) 2014, TECT. A project for the knowledge of Roman wall painting in northern Italy, Padova.
Topic 3 - TARP project.
Also in this final case the extraction of orthomosaics of excavation trenches from 3D models acquired by photogrammetry and their characterization via GIS or CAD is extremely widespread. As correctly pointed out, the acquisition of 3D models allows to store a range of additional information that can be interrogated and re-analyzed in the post-excavation phase, such as morphological details and stratigraphic relationships. From 3D models, it is also possible to extract a posteriori sections and prospectuses at specific heights/coordinates. This represents a powerful tool in the post-excavation phase. I suggest adding some other references regarding the present topic, with application to another Sardinian archaeological area such as:
Berto et al. 2021, The Photogrammetric Survey of the Phoenician and Punic Necropolis of Nora and Three-Dimensional Rendering Tools for Sharing Data, Environmental science proceedings, 10, 17. https://doi.org/10.3390/environsciproc2021010017
I would also add some general references of the 3D acquisition techniques based on photogrammetry such as:
- Remondino, F.; Campana, S. 3D Recording and Modelling in Archaeology and Cultural Heritage. Theory and Best Practices; BAR Publishing: Oxford, UK, 2014.
- Guery, J.; Hess, M.; Mathys, A. Digital Techniques for Documenting and Preserving Cultural Heritage; Arc Humanities Press: York, UK, 2017.
Finally, I detected some formatting issues:
- Line 13: “organi-zation” edit into “organization”
- Lines 86-87: why is reported her/she in these two lines referring to a group of students?
- Line 490: edit “isn’t” into “is not”
- Line 504: two full stops detected.
Finally, I suggest to edit 3-D into 3D for homogeneity throughout the text.
In conclusion, I do believe that the article deserves publication with minor revisions. The implementation of the bibliographic references about theory and other practical applications is necessary.
Author Response
I am very grateful to Reviewer 2 for their close reading of my paper and their kind work to share very useful bibliographic material. I will work to incorporate as much of it as possible. Because the context of this submission is a long-delayed publication of the proceedings of the On Shifting Ground conference in 2019, my paper reflects my remit in 1. presenting my remarks, 2. the time in which they were written, and 3. the purpose of memorializing them in this paper. Therefore, some citations that post-date 2019 might not be appropriate to include. Others may not fit the reference being made in the text, but I will make every effort to include every apt citation.
[1]= "Because the context of this submission is a long-delayed publication of the proceedings of the On Shifting Ground conference in 2019, my paper reflects my remit in 1. presenting my remarks, 2. the time in which they were written, and 3. the purpose of memorializing them in this paper. "
Reviewer 2:
"My concern mainly regards the weak bibliographic apparatus, that should be reinforced, as most part of the applications of digital tools mentioned in the text are already widely developed and commonly applied in their broad outlines to diverse archaeological projects." - I have incorportated all suggestions of bibliography as appropriate based on the age of the paper submitted (see [1]). These include:
Landeschi et al. 2015
Berggren et al. 2015
Ghedini et al. 2007.
Remondino et al. 2014.
Guery et al. 2016
"Finally, I detected some formatting issues". - I have accepted all these corrections in the revised manuscript.
Round 2
Reviewer 1 Report
I regret that two out of three suggestions were not accepted. It is fine not to add more pictures, but the paragraph structure can be greatly improved, as suggested, even though it is research work completed several years ago.